# CT++: COMPLEMENTARY CO-TRAINING FOR SEMI-SUPERVISED SEMANTIC SEGMENTATION

## ABSTRACT

With limited annotations, semi-supervised semantic segmentation aims to enhance the segmentation ability through abundant unlabeled images. Among recent trends, co-training is gaining increasing popularity, where two parallel models produce pseudo labels for each other. The success of co-training heavily relies on the discrepancy of peer models. To achieve this, prior works mostly leverage different initializations in decoders. Unfortunately, the two models still quickly converge to an extremely coupling state, making co-training downgrade to poorer self-training. To address this dilemma, we present our CT++, *decoupling* dual Co-Training models from two novel perspectives. First, we propose to construct complementary feature-level views. We design two co-training models to utilize disjoint and complementary sets of features for decoding. Apart from complementary features, we further seek complementary input views for the two models to learn respectively. Our two complementary principles enlarge the model discrepancy significantly, enabling co-training models to transfer distinct knowledge to each other and broaden their capability. This contributes to remarkably boosted co-training effectiveness. Extensive studies on Pascal, Cityscapes, COCO, and ADE20K exhibit the strong superiority of our method, *e.g.*, 80.2% mIoU with only 92 labels on Pascal.

## 1 INTRODUCTION

Semantic segmentation is a fundamental research topic in computer vision. It is indispensable for some critical applications, such as autonomous driving (Cordts et al., 2016) and medical image analysis (Ronneberger et al., 2015). Nevertheless, the bloom of previous semantic segmentation algorithms (Long et al., 2015; Zhao et al., 2017) relies heavily on the large number of labeled images, which are laborious and expensive to collect in real-world scenarios. Recently, in order to alleviate the labeling cost, semi-supervised semantic segmentation is attracting increasing attention. It only utilizes a handful of labeled images, along with abundant unlabeled images, to train a promising semantic segmentation model.

The core practice in semi-supervised semantic segmentation is to assign pseudo semantic masks to unlabeled images. It then combines manually and pseudo labeled images for joint training. Most existing works achieve this via self-training (Figure 1a), where a single model produces pseudo labels by itself and then learns in a bootstrapping manner. However, noisy pseudo labels are easy to accumulate in this process. Motivated by this, starting from CPS (Chen et al., 2021), a stream of the latest works (Fan et al., 2022; 2023; Zhang et al., 2021; Zheng et al., 2022) resort to the co-training framework (Qiao et al., 2018). In co-training (Figure 1b), dual models of the same structure but of different decoder initializations, produce pseudo labels for each other to learn. Compared with self-training, co-training can better resist the noisy pseudo labels from their

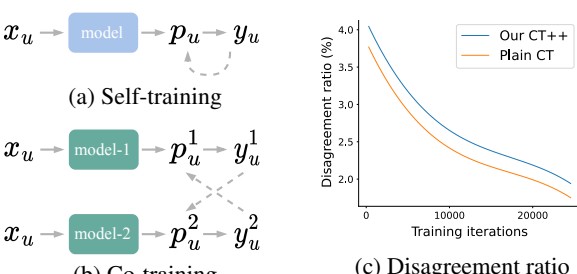

(a) Self-training

(b) Co-training

(c) Disagreement ratio

Figure 1: (c) We measure the disagreement ratio of pseudo labels between co-training models, which serve as an indicator for model discrepancy. Our CT++ consistently enlarges the discrepancy, which is essential to final co-training effectiveness (Nigam & Ghani, 2000).

peers due to model parameter discrepancy. However, we observe that as training proceeds, the co-training models still quickly converge to an extremely close state (Figure 1c "Plain CT"), making co-training downgrade to a poorer self-training method.

In this work, we highlight that the discrepancy between co-training models plays a key role in their final performance. To this end, we propose two simple yet highly effective approaches to enhancing the model discrepancy. Before introducing our methods, let us first look back at how traditional co-training methods (Nigam & Ghani, 2000) ensure the discrepancy. In most cases, there are two independent views of the same data. Therefore, each model is trained with its private view and thus can provide its peer with some unique knowledge. However, in current semantic segmentation benchmarks, there is only a single view available for an image. Thus, a question arises: *how can we obtain two meaningful and meantime not strongly correlated views from a single image?*

We observe that, although a single image *cannot* be decomposed into dual views for training, *its high-level features can*. It is acknowledged that the high-level features extracted by an ImageNet pre-trained encoder are of specific meanings (Zhou et al., 2016), *e.g.*, different feature channels correspond to varying semantic concepts. Therefore, we propose to randomly decompose the feature maps along the channel dimension into two disjoint and complementary sets. The two non-overlapping feature sets can be considered as two different yet meaningful views of an image. For example, one set of features may be sensitive to textures, while another set is responsible for the structure information. Two co-training models then forward these two complementary features into their respective decoders. The backward process will also affect the prior encoders, making both encoders and decoders (*i.e.*, the whole models) sufficiently decoupled, eventually improving the co-training effectiveness.

The aforementioned complementary strategy enlarges the discrepancy at the feature level. To further strengthen the discrepancy, we also manage to construct input-level complementary views. We are inspired by the prevalent CutMix strategy (Yun et al., 2019) in our field, which has been proven very beneficial (Chen et al., 2021; Yang et al., 2023; Zhao et al., 2023b). The basic CutMix randomly selects a rectangle region for pasting, and co-training models in recent works (Chen et al., 2021) share exactly the same processed image for training, suppressing the co-training effectiveness. Differently, we re-design a complementary CutMix version that is tailored for co-training. We switch the "copy" and "paste" roles between a pair of CutMix images to obtain two complementary input views. From these two views, our two models learn complementary knowledge and acquire different preferences. In this way, their co-training effectiveness can be significantly amplified.

To sum up, our contributions lie in three folds:

- We point out that existing co-training methods suffer from the severe prediction-coupling issue, *i.e.*, two co-training models quickly converge to similar states and produce extremely close predictions. Co-training methods gradually downgrade to poorer self-training.

- To address the above dilemma, we propose to decouple co-training models from two new perspectives. First, we construct disjoint and *complementary feature views* from a single image for the two models to decode, respectively. Moreover, the basic CutMix is modified for *complementary input views*, which are tailored for co-training.

- Integrating our two approaches, the holistic Complementary Co-Training framework CT++ enlarges the model discrepancy evidently and exhibits remarkable superiority over previous works on Pascal, Cityscapes, rarely explored COCO and ADE20K datasets. Comprehensive ablation studies also demonstrate the necessity of our complementary designs.

## 2 RELATED WORK

**Semi-Supervised Learning (SSL).** The essence of SSL is to leverage unlabeled data to facilitate the algorithm with limited labeled data. Various promising methods (Lee et al., 2013; Laine & Aila, 2017; Tarvainen & Valpola, 2017; Miyato et al., 2018; Xie et al., 2020a;b; Zoph et al., 2020; Pham et al., 2021) have been proposed in the last decade of deep learning era. Among these works, a milestone work FixMatch (Sohn et al., 2020) simplifies its pioneers (Berthelot et al., 2019; 2020), integrating two mainstream of methods, *i.e.*, pseudo labeling and consistency regularization, into a hybrid framework. Nevertheless, FixMatch only uses a single model to produce pseudo labels for itself to learn. Thus, knowledge is limited, and noisy predictions quickly accumulate.

In comparison, our CT++ is based on the co-training framework, where each model has its own preference and unique knowledge. This makes their exchanged pseudo labels informative and also enables them to be robust to potential noise from their peers.

**Semi-Supervised Semantic Segmentation.** Recent works mostly follow the trend in SSL, exploring better consistency regularization (Xie et al., 2020a). Along this trend, the augmentations CutMix (French et al., 2020) and ClassMix (Olsson et al., 2021) are demonstrated to be very important. Then ST++ (Yang et al., 2022) further proves basic color augmentations are also critical to successful consistency regularization in the multi-stage framework. Recently, UniMatch (Yang et al., 2023) has achieved substantial results by maintaining three learnable branches of image-level and feature-level augmentations. In comparison, our complementary CutMix does not aim at optimal augmentations, but to enlarge the model discrepancy. There are some other works (Alonso et al., 2021; Yuan et al., 2021; Kwon & Kwak, 2022) boosting the performance from different perspectives, such as calibrating imbalanced class distributions (He et al., 2021; Zhou et al., 2021; Hu et al., 2021; Guan et al., 2022), contrasting positive and negative pairs (Zhong et al., 2021; Lai et al., 2021; Zhou et al., 2021; Wang et al., 2022; Liu et al., 2022a), and better augmentation options (Zhao et al., 2023a;b).

The above works are all based on the self-training pipeline. Our work is orthogonal to them because we especially focus on the co-training methodology, and address how to enlarge the model discrepancy by our complementary designs. To prove this, we will present further improvements in our experiments by integrating UniMatch (Yang et al., 2023).

**Co-Training.** Ever since CPS (Chen et al., 2021) started the co-training trend in this field, considerable methods have been proposed to further boost it. For instance, UCC (Fan et al., 2022) utilizes two independent heads and re-weights pseudo labels by uncertainty. Other than prediction-level mutual supervision, TCC (Zheng et al., 2022) further proposes to enforce consistency at the feature level between two co-training models. These prior works design various auxiliary or modified losses for improvement. Our CT++ is clearly distinguished from them in that, we tackle a more essential issue in co-training, *i.e.*, how to enlarge the model discrepancy. With evidently enhanced diversity, we only need to use an extremely effortless and basic loss to achieve the best performance.

## 3 CT++: COMPLEMENTARY CO-TRAINING

In this section, we primarily introduce a basic co-training framework (Section 3.1). Based on it, we describe our two complementary designs in detail (Section 3.2 and 3.3), to enlarge the discrepancy between co-training models. An overview of our CT++ is shown in Figure 2.

### 3.1 CO-TRAINING FRAMEWORK

Co-training uses two independent models to produce pseudo labels for each other to learn. In CPS (Chen et al., 2021), these two models differ in their randomly initialized decoders, while their encoders share the same pre-trained initialization. Generally, given an unlabeled image $u$, we first produce two versions of it with weak and strong data augmentations: $u^w = \mathcal{A}^w(u)$ and $u^s = \mathcal{A}^s(\mathcal{A}^w(u))$. The $\mathcal{A}^w$ is basic spatial transformations, including horizontal flipping, resizing, and cropping. The $\mathcal{A}^s$ in CPS is only composed of CutMix (Yun et al., 2019). We diversify $\mathcal{A}^s$ with strong color transformations (Yang et al., 2023), including color jittering, blurring, and grayscaling. We will demonstrate these color transformations are rather beneficial to co-training (Figure 4). Both $u^w$ and $u^s$ are fed into two co-training models $f_1$ and $f_2$ to obtain a total of four predictions:

$$p_1^w = f_1(u^w); \; p_2^w = f_2(u^w).$$
$$p_1^s = f_1(u^s); \quad p_2^s = f_2(u^s). \tag{1}$$

Obviously, $p^w$ is of much higher quality than $p^s$, while $u^s$ is more informative to train than $u^w$. A generic practice thus is to supervise $p^s$ with $p^w$ (for simplicity: $p^w \to p^s$). In co-training, we can mutually enforce $p_1^w \to p_2^s$ and $p_2^w \to p_1^s$. Such a cross-teaching mechanism is better than self-teaching, because even if there exists noise in $p_2^w$, the other model $f_1$ can still be more resistant to it (Chen et al., 2021). Formally, the co-training loss $\mathcal{L}_{co}$ can be formulated as:

$$\mathcal{L}_{co} = \frac{1}{B_u} \sum \mathbb{1}(\max(p_1^w) \geq \tau)\mathrm{H}(p_1^w, p_2^s) + \frac{1}{B_u} \sum \mathbb{1}(\max(p_2^w) \geq \tau)\mathrm{H}(p_2^w, p_1^s), \tag{2}$$

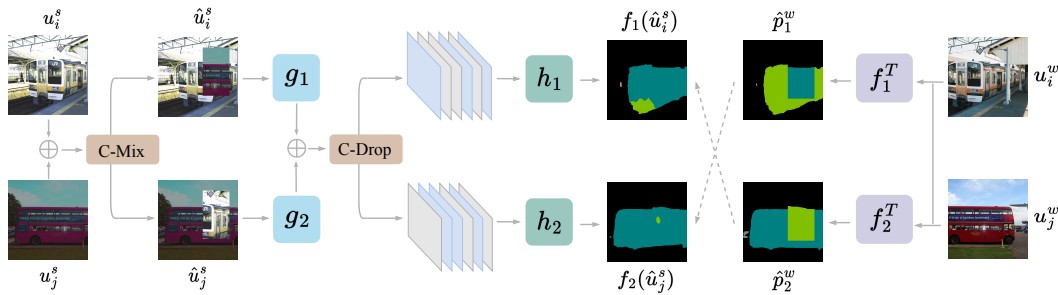

Figure 2: Illustration of CT++. A segmentation model $f$ consists of an encoder $g$ and a decoder $h$. We equip each model ($f$) with an EMA teacher ($f^T$). We present two input-level and feature-level complementary designs for larger model discrepancy. Our Complementary Dropout (C-Drop) randomly splits high-level features (extracted by $g$) into two complementary sets for each decoder ($h$) to learn. Our Complementary CutMix (C-Mix) further provides two complementary input views.

where $B_u$ is the unlabeled batch size and $\max(p_1^w)$ measures the model confidence according to maximum softmax output. The $\tau$ is the confidence threshold to select pixel-level reliable pseudo labels. The H minimizes the cross entropy between two probability distributions. In this work, we simply use the one-hot label of $p^w$ as the pseudo label.

In addition to $\mathcal{L}_{co}$, there is also a supervised cross-entropy loss $\mathcal{L}_{sup}$ for labeled images. Therefore, the overall loss is:

$$\mathcal{L} = \mathcal{L}_{sup} + \lambda_u \mathcal{L}_{co}, \tag{3}$$

where $\lambda_u$ serves as a trade-off term between the two losses, which is set as 1.0 by default.

In practice, we maintain an EMA teacher $f^T$ (Tarvainen & Valpola, 2017) with momentum 0.996 for each model to yield more stable pseudo labels. Thus, we replace aforementioned $p^w$ with $\hat{p}^w$, where $\hat{p}^w = f^T(u^w)$, as shown in the right side of Figure 2.

## 3.2 COMPLEMENTARY CHANNEL-WISE DROPOUT

The two models from the above co-training framework are still strongly coupled with each other. Such an advanced and promising co-training method gradually downgrades to a plain and weaker self-training pipeline. Inspired by conventional co-training methods, we manage to obtain two meaningful and also not strongly correlated views from a single image for two models to learn, respectively. We observe that, the high-level features extracted by the encoder can be decomposed into two complementary sets of features along the channel dimension. To enlarge the discrepancy, each co-training model uses its private collection of features for subsequent decoding.

Technically, to achieve this, we propose a Complementary Channel-Wise Dropout to acquire two disjoint and complementary sets of features. Generally, suppose a semantic segmentation $f$ is composed of an encoder $g$ (e.g., ResNet (He et al., 2016)) and a decoder $h$ (e.g., ASPP (Chen et al., 2018)). We first use $g$ to extract feature maps $F \in \mathbb{R}^{C \times H \times W}$ from a strongly augmented image $u^s$:

$$F_1 = g_1(u^s); \quad F_2 = g_2(u^s). \tag{4}$$

Then, in each training iteration, we randomly generate a dropping mask $M^F$ for channel-wise features. $M^F$ is a binary mask sampled from the binomial distribution, with half of its channels ($C/2$) all set as 1, and others as 0.

With $M^F$, we can perform complementary channel-wise Dropout on features $F_1$ and $F_2$ by:

$$\begin{aligned} \hat{F}_1 &= F_1 \odot M^F \times 2; \\ \hat{F}_2 &= F_2 \odot (1 - M^F) \times 2, \end{aligned} \tag{5}$$

where the last scaling factor 2 is to ensure the output expectation is of the same scale as normal features, which is known as "inverted Dropout".

Note that $F_1$ and $F_2$ are extracted by a pre-trained encoder. Thus, they share the same characteristics per channel. Therefore, the complementary Dropout masks $M^F$ and $(1 - M^F)$ will enable us to

obtain two disjoint sets of features with disjoint meanings. Lastly, the complementary features $\hat{F}_1$ and $\hat{F}_2$ are sent into corresponding decoders for final predictions:

$$p_1^s = h_1(\hat{F}_1); \ \ p_2^s = h_2(\hat{F}_2). \tag{6}$$

Although the complementary Dropout is inserted at the intersection of the encoder and decoder, the gradient will be back-propagated to respective encoders, making the whole model different from its peer, as shown in Figure 2. We will also discuss various dropping positions in Table 8.

### 3.3 COMPLEMENTARY SPATIAL-WISE CUTMIX

In addition to feature-level and channel-wise complementary views, we further design a Complementary CutMix to capture spatial-wise complementary views of input images. Original CPS generates a strongly augmented image $\hat{u}_i^s$ from a pair of images $u_i^s$ and $u_j^s$ with a random CutMix mask $M^I$:

$$\hat{u}_i^s = u_i^s \odot M^I + u_j^s \odot (1 - M^I). \tag{7}$$

In CPS, the obtained $\hat{u}_i^s$ is shared by two co-training models as training images. However, with precisely the same training images, the two models will be highly coupled with each other. To this end, apart from $\hat{u}_i^s$, we propose to produce a complementary CutMix view $\hat{u}_j^s$ for it:

$$\hat{u}_j^s = u_i^s \odot (1 - M^I) + u_j^s \odot M^I. \tag{8}$$

With these two complementary views (visualized in the left side of Figure 2), we can feed them into different models for training respectively. Correspondingly, the pseudo labels on weak views are also interpolated, guided by the CutMix masks:

$$\begin{aligned}
\hat{p}_1^w &= f_1^T(u_i^w) \odot (1 - M^I) + f_1^T(u_j^w) \odot M^I; \\
\hat{p}_2^w &= f_2^T(u_i^w) \odot M^I + f_2^T(u_j^w) \odot (1 - M^I).
\end{aligned} \tag{9}$$

Overall, the final unsupervised co-training loss can be formulated as:

$$\mathcal{L}_{co} = \frac{1}{B_u} \sum \mathbb{1}(\max(\hat{p}_1^w) \geq \tau) \mathrm{H}\big(\hat{p}_1^w, f_2(\hat{u}_j^s)\big) + \frac{1}{B_u} \sum \mathbb{1}(\max(\hat{p}_2^w) \geq \tau) \mathrm{H}\big(\hat{p}_2^w, f_1(\hat{u}_i^s)\big), \tag{10}$$

where complementary feature Dropout is also injected on the feedforward of $f_1(\hat{u}_i^s)$ and $f_2(\hat{u}_j^s)$.

Our holistic CT++ is depicted in Figure 2. We report the test results with the first student model $f_1$.

## 4 EXPERIMENT

### 4.1 IMPLEMENTATION DETAILS

Following previous works, we use DeepLabv3+ based on ResNet (Chen et al., 2018) as our segmentation model on Pascal (Everingham et al., 2015) and Cityscapes (Cordts et al., 2016). For faster training speed and lower GPU memory consumption, the ResNet uses an output stride of 16. On rarely explored COCO (Lin et al., 2014), we follow (Zou et al., 2021) to use DeepLabv3+ with Xception-65 (Chollet, 2017). In addition, ADE20K (Zhou et al., 2017) has never been studied in this field, so we choose to adopt the simple yet more advanced SegFormer (Xie et al., 2021) with a MiT-B4 encoder. The decoders in DeepLabv3+ and SegFormer both require multi-scale feature maps as inputs. Therefore, we randomly sample two complementary dropping masks at each scale between two co-training models. For more training and evaluation details, please refer to our appendix.

### 4.2 COMPARISON WITH STATE-OF-THE-ART METHODS

**Pascal VOC 2012** contains 21 foreground and background classes. It originally contained 1,464 training images with high-quality annotations. It was later expanded by SBD (Hariharan et al., 2011) with extra 9,118 coarsely annotated samples. There are three existing protocols to select labeled images: (a) sampling from the original high-quality training set (Table 1), (b) randomly sampling

Table 1: Results on **Pascal**. Labeled images come from the *high-quality training set*. The fraction (*e.g.*, 1/16) and number (*e.g.*, 92) in the head denote the proportion and number of labeled images. †: integrating UniMatch (Yang et al., 2023), please refer to Appendix A.3 for details. Meantime, to promote modern architectures, we also try SegFormer (Xie et al., 2021) with its MiT encoders.

| Method | Venue | Encoder | 1/16 (92) | 1/8 (183) | 1/4 (366) | 1/2 (732) | Full (1464) |
|---|---|---|---|---|---|---|---|
| Labeled Only | / | R101 | 45.1 | 55.3 | 64.8 | 69.7 | 73.5 |
| PseudoSeg (Zou et al., 2021) | ICLR'21 | R101 | 57.6 | 65.5 | 69.1 | 72.4 | 73.2 |
| ReCo (Liu et al., 2022a) | ICLR'22 | R101 | 64.8 | 72.0 | 73.1 | 74.7 | – |
| ST++ (Yang et al., 2022) | CVPR'22 | R101 | 65.2 | 71.0 | 74.6 | 77.3 | 79.1 |
| PCR (Xu et al., 2022) | NIPS'22 | R101 | 70.1 | 74.7 | 77.2 | 78.5 | 80.7 |
| GTA (Jin et al., 2022) | NIPS'22 | R101 | 70.0 | 73.2 | 75.6 | 78.4 | 80.5 |
| UniMatch (Yang et al., 2023) | CVPR'23 | R101 | 75.2 | 77.2 | 78.8 | 79.9 | 81.2 |
| iMAS (Zhao et al., 2023a) | CVPR'23 | R101 | 68.8 | 74.4 | 78.5 | 79.5 | 81.2 |
| AugSeg (Zhao et al., 2023b) | CVPR'23 | R101 | 71.1 | 75.5 | 78.8 | 80.3 | 81.4 |
| **CT++** | Ours | R101 | 75.4 | 76.9 | 78.5 | 80.0 | 81.6 |
| **CT++**[†] | Ours | R101 | **75.8** | **78.0** | **79.1** | **80.7** | **82.0** |
| **CT++** | Ours | MiT-B3 | 75.5 | 77.9 | 80.4 | 81.8 | 82.0 |
| **CT++**[†] | Ours | MiT-B3 | **77.5** | **79.4** | **82.6** | **83.9** | **84.3** |
| **CT++** | Ours | MiT-B4 | 76.2 | 78.8 | 80.6 | 82.2 | 84.1 |
| **CT++**[†] | Ours | MiT-B4 | **80.2** | **80.5** | **83.2** | **84.0** | **84.7** |

Table 2: Results on **Pascal**. Under ST++ split, labeled images are *randomly* sampled from the *blended* training set, while under U$^2$PL split, labeled images are *primarily* selected from the *high-quality* labeled set, please refer to this U$^2$PL issue for details. †: integrating UniMatch (Yang et al., 2023).

| Method | Venue | ResNet-50 | | | ResNet-101 | | |
|---|---|---|---|---|---|---|---|
| | | 1/16 (662) | 1/8 (1323) | 1/4 (2646) | 1/16 (662) | 1/8 (1323) | 1/4 (2646) |
| Labeled Only \| ST++ splits | / | 62.4 | 68.2 | 72.3 | 67.5 | 71.1 | 74.2 |
| ST++ (Yang et al., 2022) | CVPR'22 | 72.6 | 74.4 | 75.4 | 74.5 | 76.3 | 76.6 |
| PS-MT (Liu et al., 2022b) | CVPR'22 | 72.8 | 75.7 | 76.4 | 75.5 | 78.2 | 78.7 |
| UniMatch (Yang et al., 2023) | CVPR'23 | 75.8 | 76.9 | 76.8 | 78.1 | 78.4 | 79.2 |
| AugSeg (Zhao et al., 2023b) | CVPR'23 | 74.7 | 76.0 | 77.2 | 77.0 | 77.3 | 78.8 |
| **CT++** | Ours | 73.8 | 75.9 | 76.6 | 76.7 | 78.4 | 79.0 |
| **CT++**[†] | Ours | **76.0** | **77.0** | **77.5** | **78.2** | **78.6** | **79.4** |
| Labeled Only \| U$^2$PL splits | / | 67.7 | 71.9 | 74.5 | 70.6 | 75.0 | 76.5 |
| U$^2$PL (Wang et al., 2022) | CVPR'22 | 74.7 | 77.4 | 77.5 | 77.2 | 79.0 | 79.3 |
| UniMatch (Yang et al., 2023) | CVPR'23 | 78.1 | 79.0 | 79.1 | 80.9 | 81.9 | 80.4 |
| AugSeg (Zhao et al., 2023b) | CVPR'23 | 77.3 | 78.3 | 78.2 | 79.3 | 81.5 | 80.5 |
| **CT++** | Ours | 76.4 | 78.4 | 78.6 | 79.5 | 81.1 | 80.3 |
| **CT++**[†] | Ours | **78.2** | **79.6** | **79.5** | **81.1** | **81.9** | **80.9** |

from the blended 10,582 training images (Table 2 upper half), and (c) prioritizing sampling from the high-quality set, and if not enough, using the extra lower-quality training images (Table 2 lower half). As shown, with only 92 labeled images in Table 1, our CT++ outperforms iMAS (Zhao et al., 2023a) by 6.6% (68.8% → 75.4% mIoU). Moreover, CT++ achieves the best results across all settings when integrating UniMatch (Yang et al., 2023), *e.g.*, 78.0% *vs.* 77.2% (UniMatch) with 183 labels.

**Cityscapes** is a street scene dataset, consisting of 19 classes. There are 2,975 labeled training images in total. Without bells and whistles, our CT++ establishes new state-of-the-art results under many settings in Table 3. For example, with 1/4 (744) labeled images, our method boosts UniMatch by 1.1% (77.5% → 78.6%) with ResNet-50. We also report much stronger results combining UniMatch.

Table 3: Results on **Cityscapes**. †: integrating UniMatch (Yang et al., 2023), please refer to appendix.

| Method | ResNet-50 | | | | ResNet-101 | | | |
|---|---|---|---|---|---|---|---|---|
| | 1/16 (186) | 1/8 (372) | 1/4 (744) | 1/2 (1488) | 1/16 (186) | 1/8 (372) | 1/4 (744) | 1/2 (1488) |
| Labeled Only | 63.3 | 70.2 | 73.1 | 76.6 | 66.3 | 72.8 | 75.0 | 78.0 |
| CCT (Ouali et al., 2020) | 66.4 | 72.5 | 75.7 | 76.8 | 69.6 | 74.5 | 76.4 | 78.3 |
| PS-MT (Liu et al., 2022b) | – | 75.8 | 76.9 | 77.6 | – | 76.9 | 77.6 | 79.1 |
| U$^2$PL (Wang et al., 2022) | 69.0 | 73.0 | 76.3 | 78.6 | 70.3 | 74.4 | 76.5 | 79.1 |
| UniMatch (Yang et al., 2023) | 75.0 | 76.8 | 77.5 | 78.6 | 76.6 | 77.9 | 79.2 | 79.5 |
| **CT++** | 74.7 | 76.8 | 78.6 | 79.2 | 76.2 | 78.2 | 79.2 | 79.8 |
| **CT++**[†] | **76.3** | **77.2** | **78.7** | **79.3** | **77.4** | **78.9** | **79.8** | **80.2** |

Table 4: Results on **COCO**. All methods below use an Xception-65 encoder. †: integrating UniMatch.

| Method | Venue | 1/512 (232) | 1/256 (463) | 1/128 (925) | 1/64 (1849) | 1/32 (3697) |
|---|---|---|---|---|---|---|
| Labeled Only | / | 22.9 | 28.0 | 33.6 | 37.8 | 42.2 |
| PseudoSeg (Zou et al., 2021) | ICLR'21 | 29.8 | 37.1 | 39.1 | 41.8 | 43.6 |
| PC$^2$Seg (Zhong et al., 2021) | ICCV'21 | 29.9 | 37.5 | 40.1 | 43.7 | 46.1 |
| UniMatch (Yang et al., 2023) | CVPR'23 | 31.9 | 38.9 | 44.4 | 48.2 | 49.8 |
| CISC-R (Wu et al., 2023) | TPAMI'23 | 32.1 | 40.2 | 42.3 | – | – |
| **CT++** | Ours | 33.2 | 38.1 | 42.3 | 46.3 | 48.2 |
| **CT++**[†] | Ours | **34.9** | **41.1** | **45.2** | **48.7** | **50.1** |

**COCO** is a challenging benchmark for semantic segmentation, including a total of 118,287 training images. It is rarely explored in our label-efficient scenario. We follow (Zou et al., 2021) to use 80 thing classes and one background class. As shown in Table 4, across all partition protocols, our improved version CT++[†] performs consistently much better than all existing methods.

**ADE20K** has never been studied in this field. However, in view of the trend in fully-supervised scenarios and the saturated performance on Pascal and Cityscapes, we hope to make a small step toward this challenging long-tail benchmark. It is composed of 20,210 training images with a complex taxonomy of 150 classes. As demonstrated in Table 5, we first report the results with labeled images only. If using a fixed number of epochs, the total training iterations are indeed significantly reduced in low-data regimes. So, we also try to oversample limited training images to the same scale as the full training set. But this does not bring improvements. Then, after incorporating our CT++ to leverage unlabeled images, the labeled-only baseline is greatly improved across all partitions. We hope this challenging but realistic benchmark can receive more attention in future works.

## 4.3 ABLATION STUDIES

Unless otherwise specified, our ablation studies are mainly conducted on Pascal under the 183 and 732 data partitions with ResNet-101.

**Effectiveness of each component.** In Table 6, we first show the results of our co-training baseline (first row). It is much stronger than CPS (Chen et al., 2021), due to our introduced substantial color transformations (will be detailed below in Figure 4). Based on it, we further examine the effectiveness of our two main contributions, *i.e.*, complementary channel-wise Dropout (C-Drop) and complementary CutMix (C-Mix). Our two approaches both boost this strong baseline non-trivially. Moreover, by integrating both components, our CT++ achieves the best performance.

**Comparison between our complementary Dropout and random Dropout.** Although our complementary Dropout evidently enlarges the discrepancy and thus boosts the performance, we need to make sure the improvement does not come from the Dropout perturbation itself. To validate this, we perform the totally random Dropout (Srivastava et al., 2014) in each model. As shown in Figure 3a,

Table 5: Results on **ADE20K**. To promote future works with more advanced architectures, here we use SegFormer (Xie et al., 2021) with MiT-B4 as our semantic segmentation model.

| Method | 1/32 (631) | 1/16 (1263) | 1/8 (2526) | 1/4 (5052) | 1/2 (10105) |
|---|---|---|---|---|---|
| Labeled Only | 29.2 | 33.7 | 36.1 | 39.7 | 44.2 |
| + Oversample | 28.7 | 32.5 | 35.9 | 39.6 | 43.9 |
| **CT++** | **31.4 (2.2↑)** | **35.9 (2.2↑)** | **39.9 (3.8↑)** | **42.0 (2.3↑)** | **45.2 (1.0↑)** |

Table 6: Ablation studies on our complementary channel-wise Dropout (C-Drop) and complementary spatial-wise CutMix (C-Mix).

| CT++ | | Splits | |
|---|---|---|---|
| C-Drop | C-Mix | 183 | 732 |
| | | 75.7 | 79.0 |
| ✓ | | 76.6 (0.9↑) | 79.7 (0.7↑) |
| | ✓ | 76.3 (0.6↑) | 79.6 (0.6↑) |
| ✓ | ✓ | **76.9 (1.2↑)** | **80.0 (1.0↑)** |

Table 7: Advanced architectures on Cityscapes (MiT encoders (Xie et al., 2021) for Pascal are included in Table 1). We believe ViT-based models can be better if fine-tuned more carefully.

| Encoder | 1/16 (186) | 1/8 (372) | 1/4 (744) | 1/2 (1488) |
|---|---|---|---|---|
| RN-101 | 76.2 | 78.2 | 79.2 | 79.8 |
| MiT-B3 | **77.7** | 78.6 | 79.6 | 80.5 |
| MiT-B4 | 77.5 | **79.1** | **79.7** | **80.5** |

our complementary Dropout is obviously superior to the random counterpart, also indicating that enlarging discrepancy between co-training models counts.

**Comparison between our complementary CutMix and different data sequences.** Apart from our designed complementary CutMix, there is another intuitive way to feed different input views for two models. We can use two independent dataloaders for two models to sample training data, with randomly and differently shuffled data sequences. In Figure 3b, we compare these two options, observing our complementary CutMix views achieve better performance than simply shuffled views by different data sequences.

**Effectiveness of strong color transformations.** CPS (Chen et al., 2021) only uses CutMix as its strong augmentation. However, a recent trend in semi-supervised learning is to incorporate richer color transformations into the strong data augmentation pool. Therefore, based on CT++, we validate the effectiveness of our added color transformations, *i.e.*, color jittering, gaussian blurring, and grayscaling. The results in Figure 4 prove the indispensable role of vigorous color transformations.

**Position(s) of complementary Dropout.** In previous experiments, we simply performed complementary Dropout at the intersection of our encoder and decoder. We do not heavily fine-tune the optimal position to make our CT++ more general. To be more convincing, we here conduct comprehensive ablation studies on various inserted positions, as demonstrated in Table 8. It can be concluded that the most appropriate position for SegFormer is at the 3rd intermediate stage of MiT, while the ideal position for DeepLabv3+ and RN-101 is at the encoder-decoder intersection or the 3rd stage.

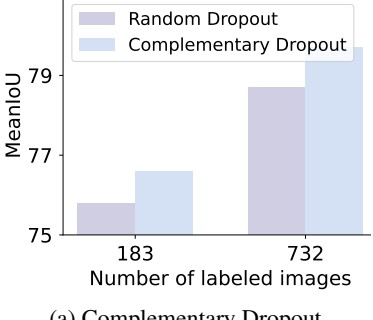

(a) Complementary Dropout

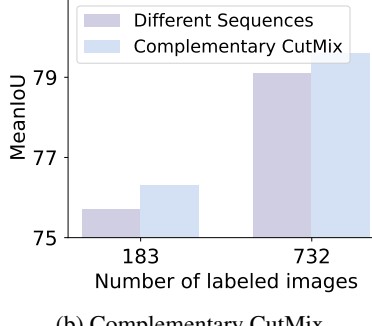

(b) Complementary CutMix

Figure 3: (a) Comparison between random Dropout and our complementary Dropout. (b) Comparison between different data sequences and our complementary CutMix.

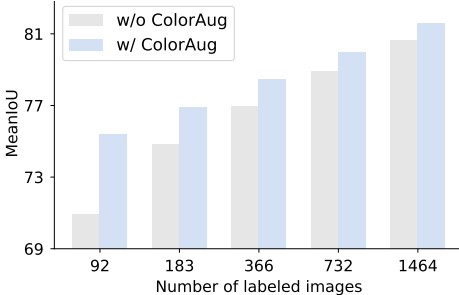
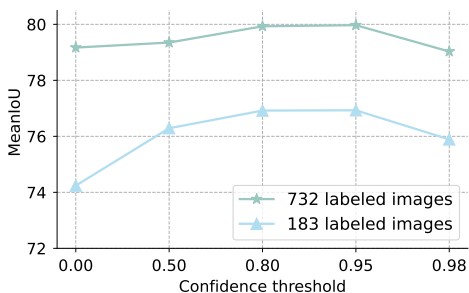

Figure 4: Effectiveness of color transformations (ColorAug). Note that, CutMix is already added.

Figure 5: Ablation studies on different values of the confidence threshold $\tau$.

Table 8: Ablation studies on various inserted position(s) of our Complementary Dropout. $N_l$ denotes the number of labeled images. We divide the encoder into four stages. Position $i$ means dropping at the output of the $i$-th stage. By default, we drop at the encoder-decoder intersection (Position 4).

| Position(s) | 4 | 3 | 2 | 1 | 3 & 4 | 2 & 4 | 1 & 4 |
|---|---|---|---|---|---|---|---|
| SegFormer MiT-B3 ($N_l = 183$) | 77.9 | **79.1** | 76.7 | 76.6 | **79.1** | 77.9 | 75.9 |
| SegFormer MiT-B3 ($N_l = 732$) | 81.8 | **82.5** | 82.5 | 81.5 | 81.6 | 81.2 | 80.6 |
| DeepLabv3+ RN-101 ($N_l = 183$) | **76.9** | 76.7 | 75.9 | 76.0 | **77.7** | 76.2 | 76.4 |
| DeepLabv3+ RN-101 ($N_l = 732$) | **80.0** | 80.1 | 79.7 | 79.1 | 79.4 | 79.1 | 79.6 |

**More advanced network architectures.** Almost all prior works in semi-supervised semantic segmentation use CNN-based models. This is not in line with the ViT trend (Dosovitskiy et al., 2020) in our community. Hence, we make some primary attempts at ViT-based models with the popular SegFormer (Xie et al., 2021). In the last four rows of Table 1 for Pascal, we tried SegFormer with MiT-B3 and MiT-B4. It requires comparable parameters (B3: 47.3M, B4: 64.0M) to DeepLab3v+ with ResNet-101 (59.4M), but achieves much better results. Moreover, in Table 7 of Cityscapes, the observation is similar. The SegFormer serials are more powerful than CNN-based models.

**Confidence threshold $\tau$.** The $\tau$ controls a trade-off between the quality and quantity of pseudo-labeled pixels. Larger $\tau$ ensures higher-quality pseudo labels, but meantime also results in fewer confident pixels. We ablate different values of $\tau$ in Figure 5. It can be observed that the widely adopted value 0.95 (Sohn et al., 2020) is already an ideal threshold for our method.

**Unsupervised loss weight $\lambda_u$.** By default, we set $\lambda_u = 1$ in all experiments, which means the supervised and unsupervised losses are of equal importance. For the universality and robustness of our CT++, we do not follow some methods (Liu et al., 2022b) to frequently change the $\lambda_u$ based on specific data partitions or datasets. However, for a more comprehensive ablation

Table 9: Ablation studies on $\lambda_u$ (1.0 by default). $N_l$ denotes the number of labeled images.

| $\lambda_u$ | 0.0 | 0.5 | 1.0 | 2.0 | 4.0 |
|---|---|---|---|---|---|
| $N_l = 183$ | 55.3 | 77.1 | 76.9 | **77.3** | 76.0 |
| $N_l = 732$ | 69.7 | 79.9 | **80.0** | 79.3 | 78.3 |

study, we attempt different values of $\lambda_u$ in Table 9. It is clear that the default value 1.0 is already promising and stable enough, and fine-tuning $\lambda_u$ may also bring some additional performance gain.

## 5 CONCLUSION

In this work, we focus on the co-training framework for semi-supervised semantic segmentation. We present two methods to enlarge the model discrepancy for better performance. Our Complementary Channel-Wise Dropout enables two models to utilize disjoint sets of feature maps for respective learning. In addition, our Complementary Spatial-Wise CutMix feeds two models with complementary input views for enlarged diversity. Our holistic co-training method, CT++, surpasses previous methods remarkably. Even more, to pursue more realistic scenarios, we also evaluate our approach on the rarely explored COCO and ADE20K datasets, and make some initial attempts at more advanced semantic segmentation architectures in this specific field.

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

# A APPENDIX

## A.1 IMPLEMENTATION DETAILS

On Pascal, each training batch is composed of 16 labeled and 16 unlabeled images, while on the other three datasets, the labeled and unlabeled batch size are both 8. The initial learning rate is set as 0.001, 0.005, 0.004, and 0.00003 for Pascal, Cityscapes, COCO, and ADE20K, respectively. For CNN-based models, we use the SGD optimizer, while for ViT-based models, we use the AdamW optimizer (Loshchilov & Hutter, 2019). The model is trained on these four datasets for 80, 240, 30, and 130 epochs, respectively. The training images are all randomly resized between 0.5 and 2.0, and then cropped to 513, 801, 513, and 512 on the four datasets respectively[1]. On Pascal and Cityscapes, according to different partition protocols, we fairly use the same data splits as ST++ (Yang et al., 2022) or U$^2$PL (Wang et al., 2022), while we follow PseudoSeg (Zou et al., 2021) splits on COCO. And on ADE20K, due to no prior works, we randomly sample labeled images by ourselves.

By default, we set the unsupervised loss weight $\lambda_u$ as 1 across all settings. And the confidence threshold $\tau$ is set as 0.95 for Pascal, COCO, and ADE20K. But on Cityscapes, we observe $\tau = 0$ is much better, which is the same as CPS (Chen et al., 2021) and UniMatch (Yang et al., 2023). Besides, on Cityscapes, we follow previous works (Chen et al., 2021; Wang et al., 2022; Yang et al., 2023) to apply online hard example mining (OHEM) (Shrivastava et al., 2016) to labeled images and use the sliding window evaluation. On the other three datasets, all images are evaluated in their original resolutions under the mean intersection-over-union metric.

## A.2 COMPUTATIONAL ANALYSIS

Compared with CPS (Chen et al., 2021), our CT++ introduces two EMA teachers. However, they indeed do not bring extra computational burden (measured by FLOPs) compared with CPS. CPS uses students to produce pseudo labels, whereas we use teachers, which does not change the FLOPs and training time. The total parameters are doubled due to extra teachers. However, the trainable parameters are the same as CPS because our teachers are in inference mode. More importantly, we highlight that we only maintain multiple models during training, while during inference, we always use a fixed student model to report the test performance, sharing exactly the same inference cost as previous methods. Below, we summarize some potentially concerned factors on computational requirements. The setting is ResNet-101 on Pascal, with a training resolution of 321.

Table 10: Comparison between CPS (Chen et al., 2021) and our CT++ on computational requirements. During inference, our CT++ enjoys the same speed and number of model parameters as previous methods, because we always use the first student model for inference.

| Method | # Params (Total / Trainable / **Test**) | FLOPs | Training Time | GPU Memory |
|--------|------------------------------------------|-------|---------------|------------|
| CPS (Chen et al., 2021) | 118.9M / 118.9M / **59.5M** | 40.3G | 1.1s / iter | 117.6GB |
| Our CT++ | 237.8M / 118.9M / **59.5M** | 40.3G | 1.1s / iter | 121.6GB |

## A.3 INCORPORATING UNIMATCH INTO CT++

In Table 1, 2, 3, and 4, we also report results of a more advanced version CT++[†], which integrates the UniMatch (Yang et al., 2023). Here, we describe the implementation details. Following the UniMatch methodology, for each co-training model, we first forward two learnable streams, each of which contains both image-level strong perturbations and our proposed complementary Dropout. The two streams differ in that the image-level strong perturbations are randomly sampled and our complementary Dropout also randomly erases some feature channels. Note that, slightly different from UniMatch, we also include feature perturbations in these image perturbation streams. because we have proved in Table 6 (first two rows) that our complementary Dropout is beneficial for image perturbation streams. In addition, similar to UniMatch, we also train our model with a third branch that contains only a random Dropout. In UniMatch, the ideal loss weight for the image-based stream and

---

[1]In Table 1, we use a smaller training size of 321 for ResNet encoders to speed up training, which is in accordance with ST++ (Yang et al., 2022) and UniMatch (Yang et al., 2023).

feature-based stream is 1:1. However, we find it is more promising to set the ratio as 7:3 if based on our CT++. We conjecture that our image-based streams also include the feature-level complementary Dropout, thus they should be attached more importance to enlarge the model discrepancy.

To better illustrate the effectiveness of our CT++ even under the strong UniMatch framework, we try to implement a basic co-training framework (also includes strong color augmentations and the EMA teachers) that incorporates UniMatch. As shown below, we find the plain co-training even deteriorates the original UniMatch performance under some settings. In comparison, our CT++ can always further boost the performance of UniMatch across all settings.

Table 11: Comparison between our CT++ and plain co-training pipeline when incorporating UniMatch. Experiments are conducted on Pascal high-quality labeled set.

| Method | 1/16 (92) | 1/8 (183) | 1/4 (366) | 1/2 (732) | Full (1464) |
|---|---|---|---|---|---|
| UniMatch (Yang et al., 2023) | 75.2 | 77.2 | 78.8 | 79.9 | 81.2 |
| UniMatch + Plain co-training | 72.8 | 77.3 | 78.2 | 80.2 | 81.3 |
| UniMatch + CT++ | **75.8** | **78.0** | **79.1** | **80.7** | **82.0** |

## A.4 QUALITATIVE RESULTS ON PASCAL VOC 2012

We provide some qualitative results on Pascal with 732 labeled images. As shown below, our CT++ performs significantly better than ST++ (Yang et al., 2022) and our labeled-only baseline in terms of semantic discrimination and boundary details.

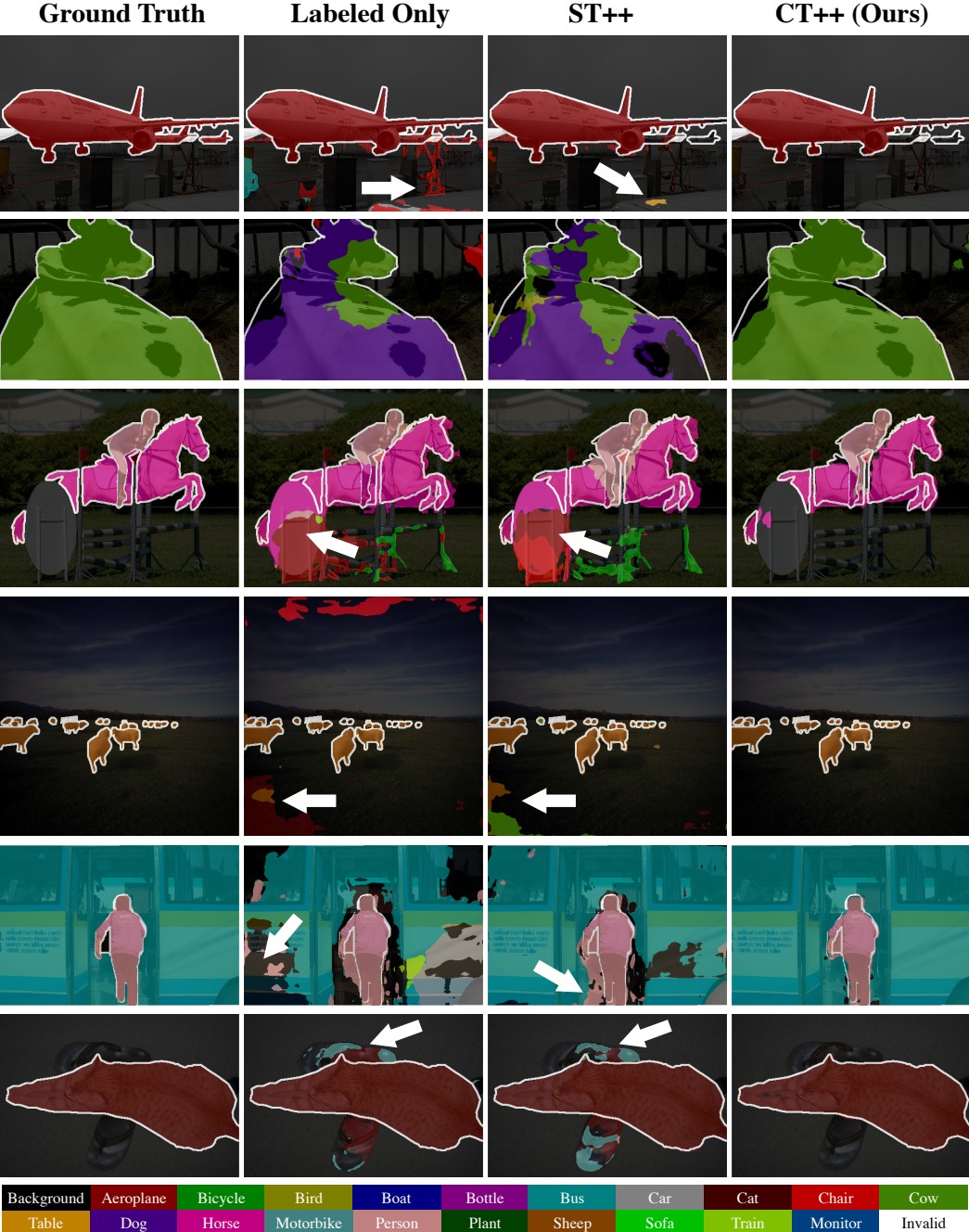

Figure 6: Qualitative results on Pascal. We compare our CT++ with ST++ (Yang et al., 2022) and the labeled-only baseline.

## A.5 QUALITATIVE RESULTS ON CITYSCAPES

We provide the qualitative results on Cityscapes with 1/4 (744) labeled images. Our CT++ is consistently much better than ST++ and the labeled-only baseline in visualization.

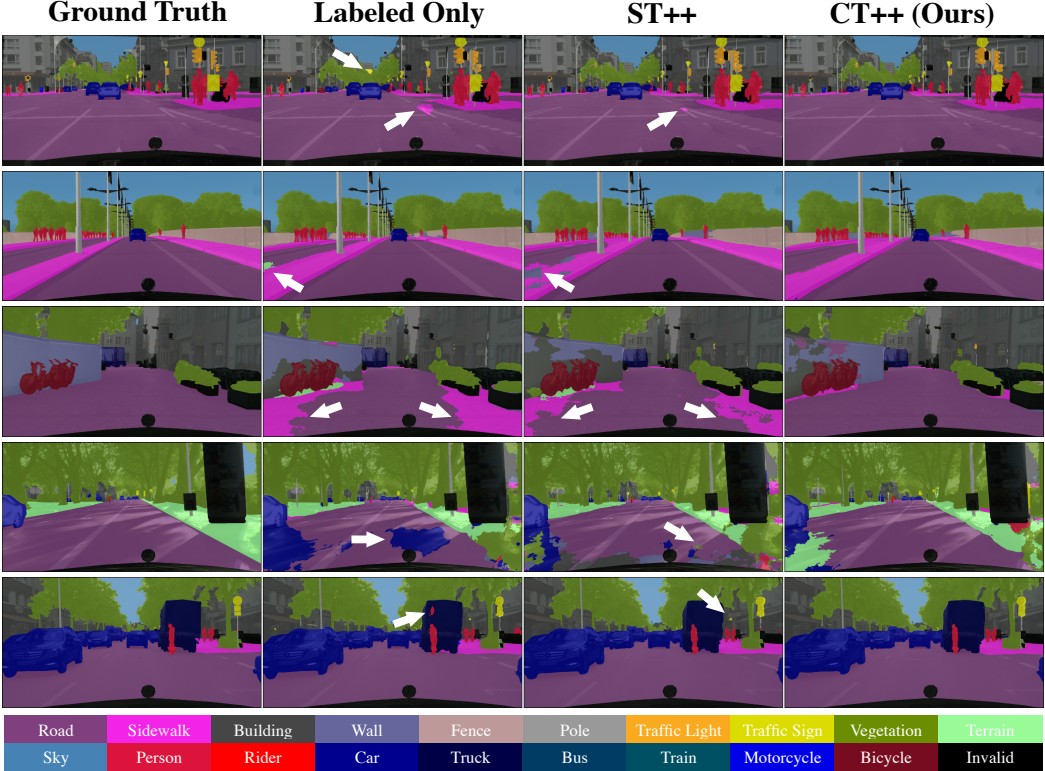

Figure 7: Qualitative results on Cityscapes. We compare our CT++ with ST++ (Yang et al., 2022) and the labeled-only baseline.

