# OpenReview forum: "CT++: Complementary Co-Training for Semi-Supervised Semantic Segmentation"
_ICLR.cc/2024/Conference — ICLR 2024 Conference Withdrawn Submission_

### Official Review · Reviewer_TKHm · 2023-10-23

**Soundness:** 3 good
**Presentation:** 3 good
**Contribution:** 1 poor
**Rating:** 3
**Confidence:** 5

**Summary:**

This paper presents an approach for performing semi-supervised semantic image segmentation. The method utilizes a co-training framework inspired by CPS [1] to generate pseudo-labels for unlabeled images. To address the issue of rapid convergence between the two models within the co-training framework, the author introduces Complementary Channel-Wise Feature Dropout, which is applied to the output features of an encoder. Additionally, the author introduces a modified form of the CutMix augmentation method called Complementary Spatial-Wise CutMix.

[1]Xiaokang Chen, Yuhui Yuan, Gang Zeng, and Jingdong Wang. Semi-supervised semantic segmentation with cross pseudo supervision. In CVPR, 2021.

[2] Sangdoo Yun, Dongyoon Han, Seong Joon Oh, Sanghyuk Chun, Junsuk Choe, and Youngjoon Yoo. Cutmix: Regularization strategy to train strong classifiers with localizable features. In ICCV, 2019.

**Strengths:**

The paper is easy to follow and well-structured. In brevity, it combines the co-training framework and channel-wise feature dropout concepts. To magnify the disparity between the two models, the author introduces both Complementary Channel-Wise Feature Dropout and a variant of the cut-mix technique. The results indicate a performance improvement on the segmentation task, including the rarely explored COCO and ADE20K datasets.

**Weaknesses:**

- Lack of technical contributions.

The primary novelty of this paper seems to lie in its approach, combining channel-wise feature dropout with a modified version of CutMix, within the framework proposed by CPS.

However, it's worth noting that the application of channel-wise feature dropout to encoder output features, subsequently fed into the decoder (the core concept of this paper), has been a prevalent technique in several recent studies [1,2,3,4] within the field of semi-supervised semantic segmentation.

The only difference is that this work uses a co-training framework (using two separate encoders) and feeds them in a complementary manner. Also, the proposed modified cut-mix algorithm seems to be a simple variant version of the original one.

So hereby the reviewer considers that this work is not enough for its novelty aspect.

[1]Y. Lu, Y. Shen, X. Xing and M. Q. . -H. Meng, "Multiple Consistency Supervision based Semi-supervised OCT Segmentation using Very Limited Annotations," *2022 International Conference on Robotics and Automation (ICRA)*, Philadelphia, PA, USA, 2022, pp. 8483-8489

[2]Dejene M. Sime, Guotai Wang, Zhi Zeng, and Bei Peng. 2023. Semi-supervised Defect Segmentation with Uncertainty-aware Pseudo-labels from Multi-branch Network. In Proceedings of the 2023 5th International Conference on Image Processing and Machine Vision (IPMV '23). Association for Computing Machinery, New York, NY, USA, 73–78.

[3]Yang, X, Tian, J, Wan, Y, Chen, M, Chen, L, Chen, J. Semi-supervised medical image segmentation via cross-guidance and feature-level consistency dual regularization schemes. *Med Phys*. 2023; 1- 13.

[4]Li, Y., Luo, L., Lin, H., Chen, H., Heng, PA. (2021). Dual-Consistency Semi-supervised Learning with Uncertainty Quantification for COVID-19 Lesion Segmentation from CT Images. In: , *et al.* Medical Image Computing and Computer Assisted Intervention – MICCAI 2021. MICCAI 2021. Lecture Notes in Computer Science(), vol 12902. Springer, Cham.

**Questions:**

According to Table 6, the author did not compare the performance of the supervised-only version (training a vanilla model only with labeled data) and a co-training (CPS version); in brief, it would be good to list these two legends in the table (Sup_only, CPS(co-training)) so that the author can figure out the proposed method's performance improvements compared to baseline more accurately.

---

### Official Review · Reviewer_RoW9 · 2023-10-24

**Soundness:** 2 fair
**Presentation:** 3 good
**Contribution:** 2 fair
**Rating:** 3
**Confidence:** 5

**Summary:**

This paper revisits co-training in semi-supervised semantic segmentation, where the key challenge is to produce two diverse models. o To ensure the discrepancy, typical theoretical co-training approaches adopt two independent views of the same data which is not available in current semantic segmentation benchmarks. To this end, the authors propose *complementary* feature and input augmentation techniques to produce disjoint views of a pair of input images. Specifically, to achieve this goal, the authors propose randomly decomposing the feature maps along the channel dimension and switching the "copy" and the "paste" roles in CutMix. Empirical evaluation on various benchmarks demonstrates the effectiveness of the proposed method.

**Strengths:**

1. This paper is well-written and easy to follow.
2. The proposed method is simple but effective.
3. Empirical evaluation is sufficient. the authors demonstrate the efficacy of CT++ on both PASCAL VOC 2012, Cityscapes, COCO, and even ADE20k datasets.

**Weaknesses:**

**Major concerns**

1. **The value of studying semi-supervised semantic segmentation.** As the Segment Anything Model [A] has been proposed and the SA-1B dataset has been made publicly available to the community, I doubt the value of this setting. The authors may argue that SAM cannot produce high-quality *semantic* masks, but several works, e.g., [B] and [C], have demonstrated that simple adaption is enough to make SAM a semantic segmentation model.

2. **The core insight has not been well ablated.** The core insight of this paper is "the success of co-training heavily relies on the discrepancy of peer models". However, except for Figure 1c, there is no other evidence to support this claim. It is better to evaluate the disagreement ratio at least for all ablations (Tables 6-8 and Figure 3). Evaluating the disagreement ratio for other methods (Tables 1-5) is also encouraged. It is strongly recommended to plot a figure that shows a strong coherence between the disagreement ratio and the mIoU.

3. **Insignificant improvements over UniMatch considering the extra computational cost.** From Table 1 and Table 2, the performance of CT++ is similar to UniMatch and the improvements are not significant. Although the improvements seem to be a little bit more significant in Table 4 and Table 5, the computational cost of CT++ is approximately 2x that of UniMatch. It is recommended to compare the number of parameters, FLOPs, training time, and GPU memory between CT++ and other self-training frameworks.

4. **The EMA teacher has not been ablated.** The improvements of CT++ over CPS might come from two extra EMA teachers.



**Minor questions/suggestions**

5. The computation of the disagreement ratio in Figure 1c needs clarifying.

6. Simply repeating the dropped feature twice as described in Eq. (5) may contribute to discrepancies between training and testing. How about replacing those masked feature channels with a learnable mask token?

7. Is the encoder for extracting F1 and F2 kept frozen?

8. It is better to list all methods in Table 1 (CPS and U2PL are missing), although CT++ performs better than those methods.



**References**

[A] A. Kirillov et al. Segment anything. arXiv preprint arXiv:2304.02643, 2023.

[B] F. Li et al. Semantic-sam: Segment and recognize anything at any granularity. arXiv preprint arXiv:2307.04767, 2023.

[C] Grounded segment anything. https://github.com/IDEA-Research/Grounded-Segment-Anything.

**Questions:**

Please refer to the weaknesses section. I really doubt the value of studying semi-supervised semantic segmentation after the segment anything model has been proposed. However, considering the good presentation of this paper, I am looking forward to an open dialog and I am willing to raise my rating if my concerns are well addressed.

---

### Official Review · Reviewer_4N2H · 2023-10-31

**Soundness:** 3 good
**Presentation:** 3 good
**Contribution:** 2 fair
**Rating:** 5
**Confidence:** 4

**Summary:**

this paper investigates the problem of semi-supervised semantic segmentation, and introduces a co-training approach for this specific task. the proposed co-training method, CT++, focuses on the disagreement of co-training models and introduces two techniques to make sure that the method can take the most from the disagreements. on multiple settings, the proposed method achieves very competitive results.

**Strengths:**

+ good results
+ easy to read
+ ablation & variant study

**Weaknesses:**

- the importance of disagreement in co-training frameworks has been pointed out by several previous studies [r1, r2]. on the other hand, some other work instead indicated that smart designs focusing only on the agreement and reducing the diversity can also improve the co-training framework [r3]. in this current manuscript, one of the major contribution is claimed to be the 'pointing out the limited disagreement'. for it to stand on its own, the reviewer would really appreciate if there is more comparison and discussion with previous work on co-training (not necessarily on semi-supervised semantic segmentation).
- the 'new' CutMix strategy in this manuscript seems a bit vanilla and its effectiveness is not fully justified. for the manuscript to claim this 'new' CutMix is actually better than previous one, it would further need experimental comparions with the existing and 'new' CutMix. as it stands, the ablation study in Table 6 only shows the removal of CutMix hurts performance, which is to be expected (since CutMix is widely adopted in semi-supervised semantic segmentation works) and cannot justify the benefit of the 'new' strategy.

overall, the reviewer feels that this manuscript provides good results (might be due to some really good implementation like mentioned in P7 baseline performance comparison, which has merit of its own). however, when it comes to the scientific contribution claimed in P2, more comparisons with existing works and experimental verification over existing strategies are needed. thus, the reviewer gives a score of borderline reject.

**Questions:**

see weakness

---

### Official Review · Reviewer_VjKK · 2023-10-31

**Soundness:** 2 fair
**Presentation:** 3 good
**Contribution:** 2 fair
**Rating:** 5
**Confidence:** 4

**Summary:**

This article works on the problem of semi-supervised semantic segmentation. It follows the co-training paradigm, but introduces several techniques (e.g., channel-wise dropout, spatial-wise cutout) to enlarge the discrepancy between co-training models. The model is verified on standard benchmarks.

**Strengths:**

The paper is very well written and easy to follow. The idea is clear and the method is simple but mostly effective.

**Weaknesses:**

My major concerns are on contributions, motivations and performance.

The contributions of the article appear to be quite incremental over CPS. The proposed techniques are more like tricks rather than contributions. For example, I will not buy that Sec. 3.3 is a major contribution.

For motivation, while the point of model discrepancy somewhat makes sense to me, I am confusing whether larger discrepancy indicates better performance. All the techniques in the paper are designed to enhance the differences between models, but to what extent we can do this. The submission lacks proper discussions regarding this aspect.

For performance, it seems from Fig. 4 that ColorAug affects a lot on performance. For example, for the case of 92 samples, we see an increase of ~4% mIoU. This raised an issue about the effectiveness of the method, does the performance improvement come from the proposed components, or the ColorAug trick?

**Questions:**

How is the disagreement computed for Fig. 1(c)? btw, for captions of Fig. 1, it would be better to also mention (a) and (b), rather than starting directly with (c).